# A toxin-antidote selfish element increases fitness of its host

Lijiang Long[1,2], Wen Xu[1], Francisco Valencia[1], Annalise B Paaby[1]*, Patrick T McGrath[1,3]*

[1]School of Biological Sciences, Georgia Institute of Technology, Atlanta, United States; [2]Interdisciplinary Graduate Program in Quantitative Biosciences, Georgia Institute of Technology, Atlanta, United States; [3]School of Physics, Georgia Institute of Technology, Atlanta, United States

**Abstract** Selfish genetic elements can promote their transmission at the expense of individual survival, creating conflict between the element and the rest of the genome. Recently, a large number of toxin-antidote (TA) post-segregation distorters have been identified in non-obligate outcrossing nematodes. Their origin and the evolutionary forces that keep them at intermediate population frequencies are poorly understood. Here, we study a TA element in *Caenorhabditis elegans* called *zeel-1;peel-1*. Two major haplotypes of this locus, with and without the selfish element, segregate in *C. elegans*. We evaluate the fitness consequences of the *zeel-1;peel-1* element outside of its role in gene drive in non-outcrossing animals and demonstrate that loss of the toxin *peel-1* decreased fitness of hermaphrodites and resulted in reductions in fecundity and body size. These findings suggest a biological role for *peel-1* beyond toxin lethality. This work demonstrates that a TA element can provide a fitness benefit to its hosts either during their initial evolution or by being co-opted by the animals following their selfish spread. These findings guide our understanding on how TA elements can remain in a population where gene drive is minimized, helping resolve the mystery of prevalent TA elements in selfing animals.

*For correspondence: paaby@gatech.edu (ABP); patrick.mcgrath@biology.gatech. edu (PTMcG)

**Competing interest:** The authors declare that no competing interests exist.

## Editor's evaluation

This important work addresses how a selfish genetic element is maintained at intermediate frequencies in *C. elegans*. The evidence is convincing with both experimental and theoretical findings that tell us more about how these elements affect transmission in populations. Overall, the results of this study will be of broad interest to evolutionary biologists.

## Introduction

Selfish genetic elements, or selfish genes, are heritable segments of DNA that promote their own transmission relative to the rest of the genome, potentially at the expense of the individual organism (*Werren, 2011*; *Werren et al., 1988*). They act through a diverse catalog of molecular mechanisms to increase their frequency, including transposons, homing endonucleases, sex-ratio distorters, and segregation or post-segregation distorters (*Hurst and Werren, 2001*). Because selfish genetic elements induce tension between genes and the hosts that carry them, including causing disease and other health problems, their discovery and study over the last 50 or so years have motivated major questions—and debate—over the nature and consequences of genetic conflict in inheritance systems (*Ågren, 2016*; *Ågren and Clark, 2018*; *Hurst and Werren, 2001*). In an early review, and in its revisit 23 years later, *Werren, 2011* posed three questions about selfish genetic elements that

remain outstanding today: (i) how they arise, (ii) how they are maintained, and (iii) how they influence evolution.

Theory and observation have indicated that selfish genetic elements decrease in prevalence as inbreeding in a system increases; spreading necessarily requires outcrossing to a vulnerable genetic background (*Ågren and Clark, 2018*; *Hurst and Werren, 2001*). However, a recent wave of discovery of toxin-antidote (TA) elements in non-obligate outcrossing species (e.g. *Ben-David et al., 2017*; *Ben-David et al., 2021*; *Noble et al., 2021*; *Nuckolls et al., 2017*; *Shen et al., 2017*) challenges this view. TA elements are post-segregation distorters composed of two or more linked sub-elements, including a 'toxin' transmitted cytoplasmically from the parent to the offspring through the gamete and an 'antidote' that rescues when expressed in the zygote. TA elements induce heavy fitness costs to hybrids heterozygous for an active/inactive genotype because while all gametes will carry the cytoplasmic toxin, only those zygotes that inherit the TA allele will express the antidote and survive.

TA systems, which include Medea elements (e.g. *Beeman et al., 1992*; *Noble et al., 2021*) and functionally similar 'gamete killers' (e.g. *Nuckolls et al., 2017*), have been identified across multiple kingdoms of life, including bacteria, plants, fungi, insects, and nematodes (*Akarsu et al., 2019*; *Bardaji et al., 2019*; *Beckmann et al., 2017*; *Beeman et al., 1992*; *Ben-David et al., 2021*; *Chen et al., 2008*; *Leplae et al., 2011*; *Saavedra De Bast et al., 2008*; *Seidel et al., 2011*; *Yang et al., 2012*). In the nematode genus *Caenorhabditis*, androdioecy (male and hermaphrodite sexes) has evolved independently three times from a male–female ancestor (*Ellis, 2017*); consequently, *C. elegans*, *C. briggsae,* and *C. tropicalis* reproduce primarily by selfing, with infrequent instances of outcrossing via male mating (*Barrière and Félix, 2005*; *Cutter et al., 2006*; *Noble et al., 2021*). Medea TA elements have been identified in all three species, including multiple elements in both *C. elegans* and *C. tropicalis* (*Ben-David et al., 2017*; *Ben-David et al., 2021*; *Noble et al., 2021*; *Seidel et al., 2008*; *Seidel et al., 2011*). These results beg the question: Why have so many TA elements been identified in non-obligate outcrossing species (*Noble et al., 2021*; *Sweigart et al., 2019*)?

One of the most complete mechanistic descriptions of a TA system is the *zeel-1;peel-1* locus in *C. elegans*, in which a sperm-delivered toxin (*peel-1*) induces arrest in embryos not carrying the zygotically expressed antidote (*zeel-1*) (*Figure 1A*; *Seidel et al., 2008*; *Seidel et al., 2011*). The alternative active/inactive haplotypes that segregate within *C. elegans* exhibit high genetic diversity (*Figure 1B*) that dates the divergence of the two haplotypes to roughly 8 million generations ago (*Lee et al., 2021*). Maintenance (*Figure 1C*) of ancient polymorphism is inconsistent with a history of selfish activity: in outcrossing populations, genic drive should fix the active haplotype rapidly; in the androdioecious mating system of *C. elegans*, a high rate of selfing should fix an element at high frequency or allow it to be lost by drift at low frequency (*Noble et al., 2021*). However, it is unknown how the fitness of a TA element, independent of its selfishness, may influence its spread or maintenance.

In this study, we investigate the fitness effect of a TA element in the host genotype, independent of its toxic incompatibility in outcrossed individuals, to assess its role in maintaining the prevalence of TA elements in non-obligate outcrossing populations. Modeling under expected conditions shows that TA elements are vulnerable to being lost at low frequency, but direct tests of fitness-proximal traits indicate that the active *peel-1* allele increases fitness relative to the inactive haplotype. These results suggest that the spread of the *zeel-1;peel-1* allele within *C. elegans* might not be gene drive, but positive selection acting on independent biological traits. These findings have consequences for considering the origin and maintenance of TA elements and their influence on the historical evolution of populations.

## Results and discussion
### The fitness cost of a TA element influences its initial spread and final fate

The effectiveness of a gene drive system is dependent on multiple factors beyond its selfish induction of incompatibility, including genotype frequency, outcrossing rate, and fitness in the host background. To explore these parameters, we adapted a family-based model (*Figure 1D*, *Table 1*; *Wade and Beeman, 1994*) with modifications to account for paternal delivery of the toxin, selfing versus outcrossing rate, and selection cost of the element.

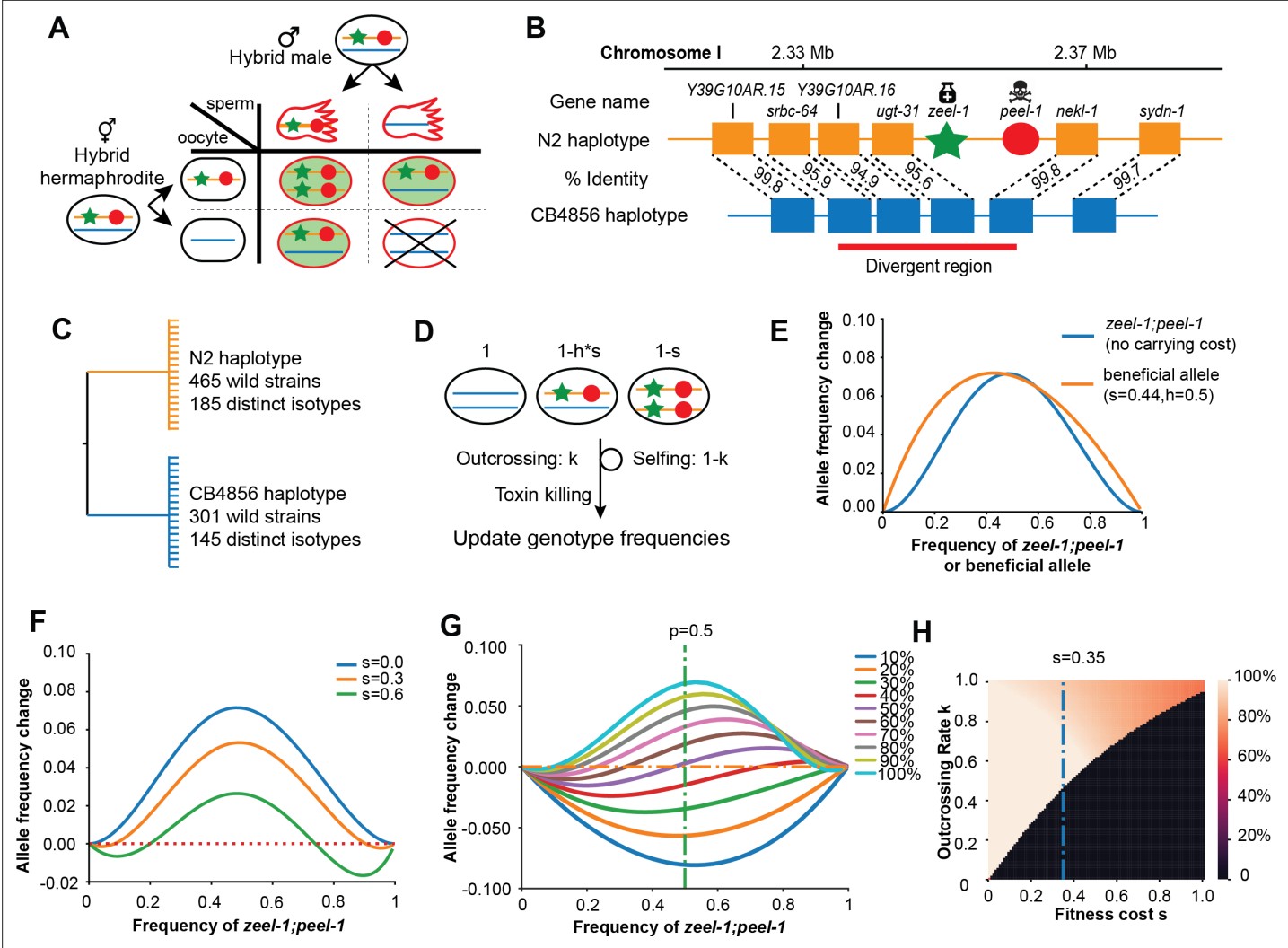

**Figure 1.** Description and models of selection for *zeel-1;peel-1*. (**A**) Schematic of the progenies created from an F1 hybrid cross, produced through intercrossing. Red outline indicates cytoplasmic inheritance of the PEEL-1 toxin from the hybrid male, independent of genomic inheritance of *peel-1* (red circle) or *zeel-1* (green star), which counteracts the toxin by zygotic expression (green background). Progeny that die are indicated by the X cross. (**B**) Schematic of the genomic region surrounding *zeel-1;peel-1* for two major haplotypes, N2 and CB4856. *zeel-1;peel-1* is present in the N2 genome and deleted in the CB4856 genome. Amino acid identities of each gene are shown between the two haplotypes. The red bar denotes the hyperdivergent region starting in the 5' end of *srbc-64* and ending in the beginning of *nekl-1*. (**C**) A gene tree representation of the *zeel-1;peel-1* locus from wild strains of *C. elegans* using the hyperdivergent region (based on ***Seidel et al., 2008***). Two major branches distinguish the N2 and CB4856 haplotypes; the number of wild isolates and distinct isotypes are labeled on each branch. This distribution is consistent with balancing selection acting on each haplotype. (**D**) Schematic of the simulation of *zeel-1;peel-1* population dynamics. The fitness of each genotype is shown on top. Genotype frequencies are updated each generation using ***Table 1***. (**E**) The allele frequency change per generation (y-axis) of *zeel-1;peel-1* (s = 0, k = 1, blue curve) or a beneficial allele (s = 0.44, h = 0.5) as a function of allele frequency (x-axis). (**F**) The change in allele frequency per generation (y-axis) of *zeel-1;peel-1* with three different carrying costs (s = 0, s = 0.3, and s = 0.6), as a function of allele frequency (x-axis). (**G**) The change in allele frequency per generation (y-axis) of *zeel-1;peel-1* with a fixed fitness cost (s = 0.35, h = 0.5) at different rates of outcrossing, as a function of allele frequency (x-axis). (**H**) Heatmap showing the *zeel-1;peel-1* frequency after 1000 generations, over varying outcrossing rates (y-axis) and carrying costs (x-axis). Initial frequency of the element was 50%. Black indicates animals that have lost the element.

The online version of this article includes the following source data and figure supplement(s) for figure 1:

**Source data 1.** Excel file containing source data for ***Figure 1***.

**Figure supplement 1.** Heatmap of *zeel-1;peel-1* frequency after 100 generations.

Under a simple scenario of no fitness consequence to the host genotype (s = 0) and a completely outcrossing population (k = 1), the element spreads rapidly through the population with a maximum allele change comparable to an additive beneficial allele with a selection coefficient of 0.44 (*Figure 1E*), 2–4 times higher than the selection coefficient of lactase persistence in humans (*Bersaglieri et al., 2004*). However, gene drive is weaker than the beneficial allele at the tails of the allele frequency range: at low frequency, the rarity of the element limits how fast it spreads; at high frequency, the rarity of the vulnerable genotype slows its approach to fixation. If the element induces a carrying cost to the host genotype (e.g. s = 0.3, s = 0.6), for example, via energy expenditure or 'leaky' toxicity, the dynamics at the extreme allele frequencies are amplified (*Figure 1F*). At low frequency, the carrying cost counteracts gene drive, reducing the likelihood that the element reaches appreciable frequency by genetic drift before being lost. At high frequency, the carrying cost compounds the slowing rate of gene drive such that it reaches a stable equilibrium and does not fix.

Previous models have shown that spread of a TA element accelerates with the rate of outcrossing (*Noble et al., 2021*). Given a substantial carrying cost to the host genotype (s = 0.35), a TA element is likely to increase in frequency only under relatively high rates of outcrossing (*Figure 1G*). Under outcrossing rates (~1%) typical for *C. elegans* (*Barrière and Félix, 2005*; *Frézal and Félix, 2015*), the element will likely be lost from the population under all but the mildest carrying costs (0.008) (*Figure 1—figure supplement 1*), as increasing fitness costs require increasing outcrossing for the element to reach a stable equilibrium (*Figure 1H*).

Given these dynamics, we are challenged to explain how a novel TA element could rise in initial frequency in a population. One hypothesis is that TA elements in non-obligate outcrossing *Caenorhabditis* may have originated in an outcrossing ancestor, then persisted by other evolutionary forces such as drift or balancing selection (*Noble et al., 2021*; *Seidel et al., 2011*; *Sweigart et al., 2019*). Such a scenario is consistent with the recent opinion by *Sweigart et al., 2019*, who argue that TA elements may exist in nature with only incidental instances of 'selfish' activity. This shift away from the conventional framing of TA elements as consistently selfish makes sense in the context of non-obligate outcrossing populations, which permit elements to proliferate in sequestered lineages without conflict.

## The active *zeel-1;peel-1* haplotype is associated with higher fitness in laboratory environments

To investigate its potential to spread through the population without conflict, we evaluated the fitness consequences of the *zeel-1;peel-1* element independent of its incompatibility cost in heterozygotes. First, we employed a previously described fitness assay (*Large et al., 2016*; *Zhao et al., 2018*) to compete N2$^{zeel-1;peel-1(CB4856)}$, which carries an ~140–370 kb interval spanning the *zeel-1;peel-1* locus from CB4856 introgressed into N2 (*Ben-David et al., 2017*), against N2$^{marker}$, a modified version of N2 carrying a silent marker mutation in the *dpy-10* gene. As CB4856 harbors the inactive haplotype, N2$^{zeel-1;peel-1(CB4856)}$ lacks the toxin/antidote element, while N2$^{marker}$ carries the active element native to N2. In these assays, males are not present and outcrossing is prevented, so relative fitness is estimated from true-breeding hermaphrodite genotypes. As a positive control, we used the N2$^{glb-5;npr-1(CB4856)}$ near-isogenic lines (NILs) strain, which carries ancestral alleles of *glb-5* and *npr-1* that decrease fitness of animals in laboratory conditions (*Zhao et al., 2018*).

N2$^{marker}$ outcompeted N2$^{zeel-1;peel-1(CB4856)}$ (*Figure 2A*), with a relative fitness (w) of 1.18 (1.15–1.21, 95% CI). Association of the active allele with higher fitness suggests that induction of *peel-1* toxicity and/or rescue by *zeel-1* is not costly, that the active allele is linked to one or more mutations in the N2 background that confer an independent fitness advantage, or both. These mutations could reside within *zeel-1;peel-1*, within the four nearby genes within the high-diversity region, or outside the high-diversity region but within the 140–370 kb introgressed region of this strain (*Figure 1B*). We also measured fecundity and body size in N2 and N2$^{zeel-1;peel-1(CB4856)}$ directly and observed similar outcomes: N2 laid 9% more embryos (p<0.001, *Figure 2B*) and was 9% larger 72 hr after hatching (p<0.001, *Figure 2C*), indicating animals grew faster, resulting in a larger body size at a similar time in development.

These results indicate that variants associated with the active *zeel-1;peel-1* haplotype promote fitness in the host genotype, providing a potential mechanism for proliferation and persistence of the element in selfing lineages.

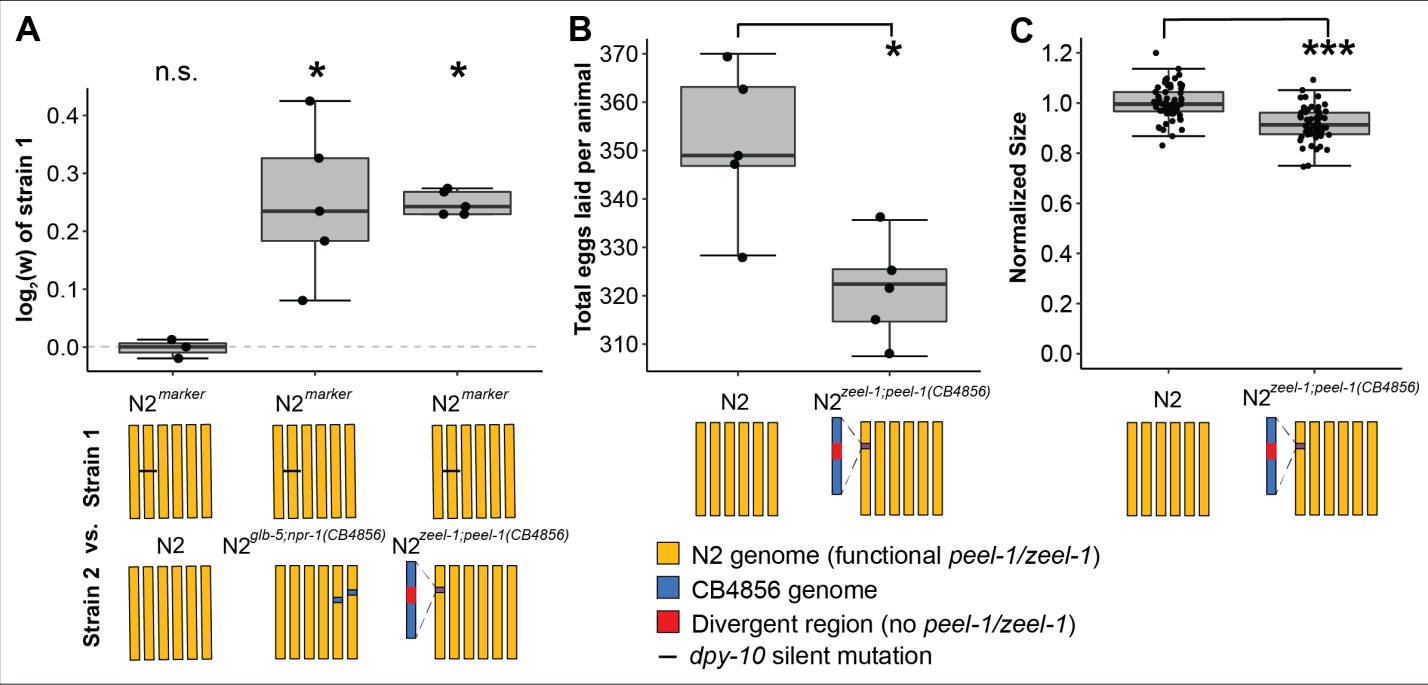

**Figure 2.** *zeel-1;peel-1* is linked to genetic variation that increases fitness in the host genotype in laboratory conditions. (**A**) Relative fitness of experimental genotypes competed against N2$^{marker}$, which has a silent mutation in *dpy-10* used as a barcode for digital PCR. This mutation exhibits no fitness effect as there was no significant difference in the competition between N2$^{marker}$ and N2. N2$^{marker}$, which has the *zeel-1;peel-1* element native to N2, outcompeted N2$^{zeel-1;peel-1(CB4856)}$, which has an ~140–370 kb interval spanning the *zeel-1;peel-1* locus from CB4856 introgressed into N2 (*Ben-David et al., 2017*). The relative fitness of N2$^{marker}$ over N2$^{zeel-1;peel-1(CB4856)}$ (w = 1.18, 1.15–1.21, 95% CI) is similar to its relative fitness over N2$^{glb-5;npr-1(CB4856)}$ (w = 1.19, 1.10–1.28, 95% CI), which was used as a positive control. N2$^{glb-5;npr-1(CB4856)}$ carries introgressed CB4856 alleles at *npr-1* and *glb-5* that were previously shown to decrease fitness relative to N2 alleles in laboratory conditions (*McGrath et al., 2009*). The N2 vs. N2$^{marker}$ and N2 $^{marker}$ vs. N2$^{zeel-1;peel-1(CB4856)}$ are identical to the data in *Figure 3E* as the competition were done (*McGrath et al., 2009*) in parallel. (**B**) Fecundity of N2 and N2$^{zeel-1;peel-1(CB4856)}$. (**C**) Growth/ size analysis of N2 and N2$^{zeel-1;peel-1(CB4856)}$. The body size of young adult animals was measured at 72 hr and normalized to the average size of N2. The N2 data for (**B**) and (**C**) is identical to the data in *Figure 3C and D*, as all three strains were analyzed on the same day. Box plots show the central 50% of the dataset and the whiskers indicate 1.5× of the interquartile range; ***p<0.001 and *p<0.05 by non-parametric analysis with correction for multiple tests (see 'Methods').

The online version of this article includes the following source data for figure 2:

**Source data 1.** Excel file containing source data for *Figure 2*.

## The active *peel-1* allele is associated with higher fitness in laboratory environments

To test the fitness consequences of the *peel-1* toxin directly, we used CRISPR/Cas9 to engineer a knockout of *peel-1* (*kah126*, or *peel-1(trunc)*) in the N2 background. N2$^{peel-1(trunc)}$ produces a truncated protein of 46 amino acids (relative to 174) via an early stop codon (*Figure 3A*). We verified loss of function by embryo killing assays: N2 crossed to CB4856 produced the expected 25% embryonic lethality from selfed F1 hermaphrodites; the N2$^{peel-1(trunc)}$ cross produced zero dead embryos (*Figure 3B*). Interestingly, the *peel-1(trunc)* allele affected fitness proximal traits and fitness in laboratory conditions. The N2$^{peel-1(trunc)}$ produced 6% fewer offspring (*Figure 3C*) and were 7% smaller 72 hr after hatching than N2 (*Figure 3D*). Competition experiments between N2$^{peel-1(trunc)}$ against N2$^{marker}$ also demonstrated a fitness increase associated with the active *peel-1* allele (w = 1.06, 1.04–1.07, 95% CI) (*Figure 3E*); this fitness difference accounts for 32% of the difference arising from the N2$^{zeel-1;peel-1(CB4856)}$ comparison. Thus, while *peel-1* acts as a toxin in the context of outcrossing cross-progeny, it increases the fitness of selfing hermaphrodites in laboratory conditions. These results suggest that *peel-1* is not simply a toxin gene and plays some other biologically relevant role in *C. elegans*. The fitness differences may be mediated via egg-laying rate. The higher total fecundity suggests that the number of self-sperm produced differs among strains, which would also affect the earliest timepoint eggs may be laid; additional experiments are needed for confirmation. These results also suggest that additional genetic

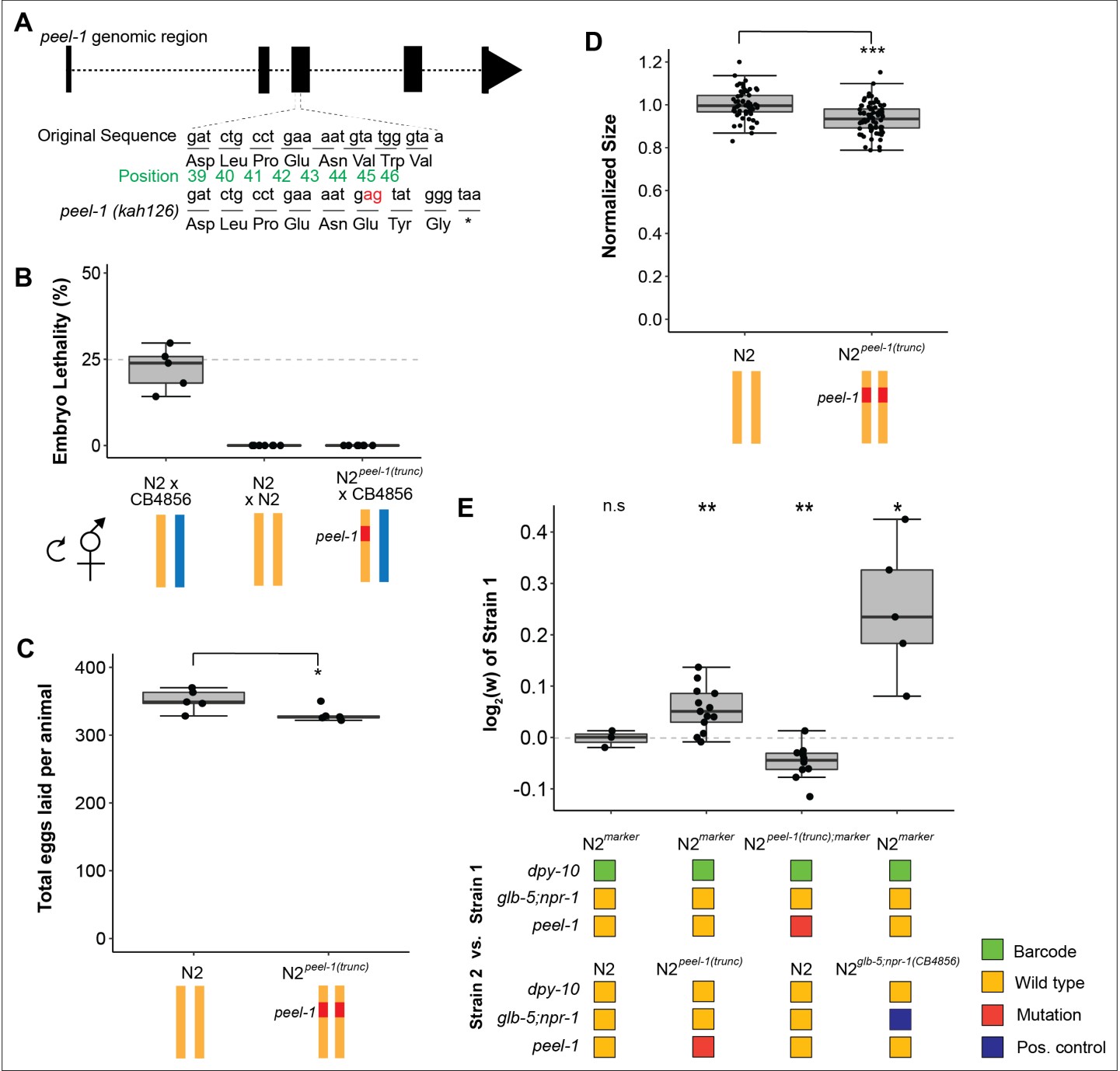

**Figure 3.** Tests of *peel-1* function using CRISPR/Cas9 show the active *peel-1* allele increases fitness. (**A**) Schematic of the *peel-1* loss-of-function allele, N2*peel-1(trunc)*. At *peel-1*, two additional nucleotides (marked in red) inserted into the third exon generate a frameshift and an early stop codon (marked by *). The green numbers denote the amino acid position of the PEEL-1 protein sequence. (**B**) N2*peel-1(trunc)* has lost *peel-1* function, as selfed cross-progeny show. As expected, N2 × CB4856 produce ~25% embryonic lethality (p=0.44 compared to null expectation of 25%), and N2 × N2 produce zero dead embryos. N2*peel-1(trunc)* × CB4856 also produce zero dead embryos, indicating loss of toxicity. (**C**) Fecundity of the N2 and N2*peel-1(trunc)* strains. (**D**) Growth/ size analysis of N2 and N2*peel-1(trunc)*. The body size of young adult animals was measured at 72 hr and normalized to the average size of N2. For (**B**– **D**), yellow represents the N2 genome, blue represents the CB4856 genome, and red represents the truncated allele of *peel-1* on chromosome I. The N2 data for (**C**) and (**D**) is identical to the data in *Figure 2B and C*, as all three strains were analyzed on the same day. (**E**) Competition assays between strains in standard laboratory conditions; positive values indicate strain 1 is more fit and negative values indicate strain 2 is more fit. Competition between the wild-type N2 *peel-1* allele and the *peel-1* loss-of-function mutation indicate a fitness benefit for *peel-1* (in assays with the marker in both backgrounds), which accounts for 32% of the difference arising from the relative fitness of the CB4856 introgression of *zeel-1;peel-1*. The N2 vs. N2*marker* and N2 *marker* vs. N2*zeel-1;peel-1(CB4856)* are identical to the data in *Figure 2A* as the competition were done in parallel. The relative fitness of N2*glb-5;npr-1 (CB4856)*

*Figure 3 continued on next page*

*Figure 3 continued*

over N2$^{marker}$ is shown as a positive control. Box plots show the central 50% of the dataset and the whiskers indicate 1.5× of the interquartile range; ***p<0.001, **p<0.01, and *p<0.05 by non-parametric analysis with correction for multiple tests (see 'Methods').

The online version of this article includes the following source data for figure 3:

**Source data 1.** Excel file containing source data for *Figure 3*.

variations linked to the *zeel-1;peel-1* locus play a role in laboratory fitness as the *peel-1* mutations did not fully phenocopy the fitness of the *peel-1* NIL line.

Our work indicates that *peel-1* plays an additional biological role outside of its role as a selfish element. Since the experiments on *peel-1* relied on a single CRISPR/Cas9-generated strain, we were worried that background mutations could account for the differences in fitness and fitness-proximal traits of this strain. To address this, we generated six additional alleles modifying *peel-1* (*Figure 4*). First, we created two replicate alleles (*kah1000* and *kah1001*) that revert the original *peel-1* mutant allele (*kah126*) back to wild-type. Second, we created three replicate alleles (*kah1003-5*) with an edit in the third codon to induce an early stop. Finally, we created an allele (*kah1006*) with a 5 bp deletion that excised the ATG start codon. Unlike the original N2$^{peel-1(trunc)}$ strain, with a stop codon in the third exon that could potentially lead to a truncated protein product, these latter four strains are predicted to create true null alleles that should prevent the production of any *peel-1* protein. We verified *peel-1* activity using embryo killing assays on one strain of each allele type; as expected, self-progeny of the heterozygous offspring of the strain with the reversion allele of *peel-1* showed 25% lethality, and self-progeny from crosses with the loss-of-function alleles of *peel-1* showed zero lethality (*Figure 4C*). We tested these strains in competition experiments and observed equivalent performance among the replicate genotypes within the reversion and early stop allele classes, and further, equivalent performance among all *peel-1* mutants (*Figure 4D*). As expected, strains carrying the reversion alleles showed no significant difference in fitness in competition with wildtype, while strains carrying the new loss-of-function alleles, like the original N2$^{peel-1(trunc)}$ strain, were significantly outcompeted (*Figure 4D*). These experiments strongly support a role for *peel-1* outside of its role as a selfish element. We did not test these additional strains to confirm that the egg-laying and growth rate phenotypes that we measured in the original *peel-1* loss-of-function, so we cannot exclude the possibility that this phenotypic difference is due to background mutations.

This is not necessarily surprising, as the role of *peel-1* in a secondary biological process was considered in its initial characterization (*Seidel et al., 2011*). Such a role would help the initial spread of the element during its formation, when its low frequency (where gene drive is ineffective) and its initial toxicity (before *zeel-1* could evolve to counteract it) should prevent its spread. Our work supports that model, suggesting that both roles of *peel-1* could co-evolve together. But then, why has not the element fixed? The *zeel-1;peel-1* locus shows a signature of balancing selection, which appears widespread in *C. elegans*. Hyperdivergent regions, including that spanning *zeel-1;peel-1*, punctuate the genome; balancing selection across diverse ecological niches may explain their maintenance (*Lee et al., 2021*). Previously, maintenance of the *zeel-1;peel-1* element was hypothesized to arise from tight linkage to a nearby polymorphism under balancing selection (*Seidel et al., 2008*). Our results suggest that *peel-1* could be under balancing selection itself. *peel-1* confers a fitness benefit within the lab environment, and it may pleiotropically influence other life history traits or affect fecundity and growth rate differently in different environments, providing alternate fitness strategies for local adaptation.

Previous work has suggested that TA elements may shape evolution by promoting selfing to escape the cost of selfish gene drive (*Noble et al., 2021*). Here we provide a mechanism for their spread and maintenance that helps to explain their prevalence in selfing *Caenorhabditis* (*Ben-David et al., 2021*; *Noble et al., 2021*; *Sweigart et al., 2019*). Moreover, our observation of a toxin directly affecting biological traits mirrors work in transposable elements, which are also selfish elements that can be domesticated for phenotypic benefit to the organism (*Werren, 2011*). This previously undescribed, non-toxin related role of a TA element is expected to shape evolutionary trajectories of both the element and the organism.

In the future, it will be interesting to parse the mechanism by which *peel-1* affects fitness in the adult hermaphrodite. *peel-1* transcripts are restricted to sperm (*Seidel et al., 2011*), but these results suggest that it can affect adult phenotypes like fecundity and growth rate. One possibility is that the

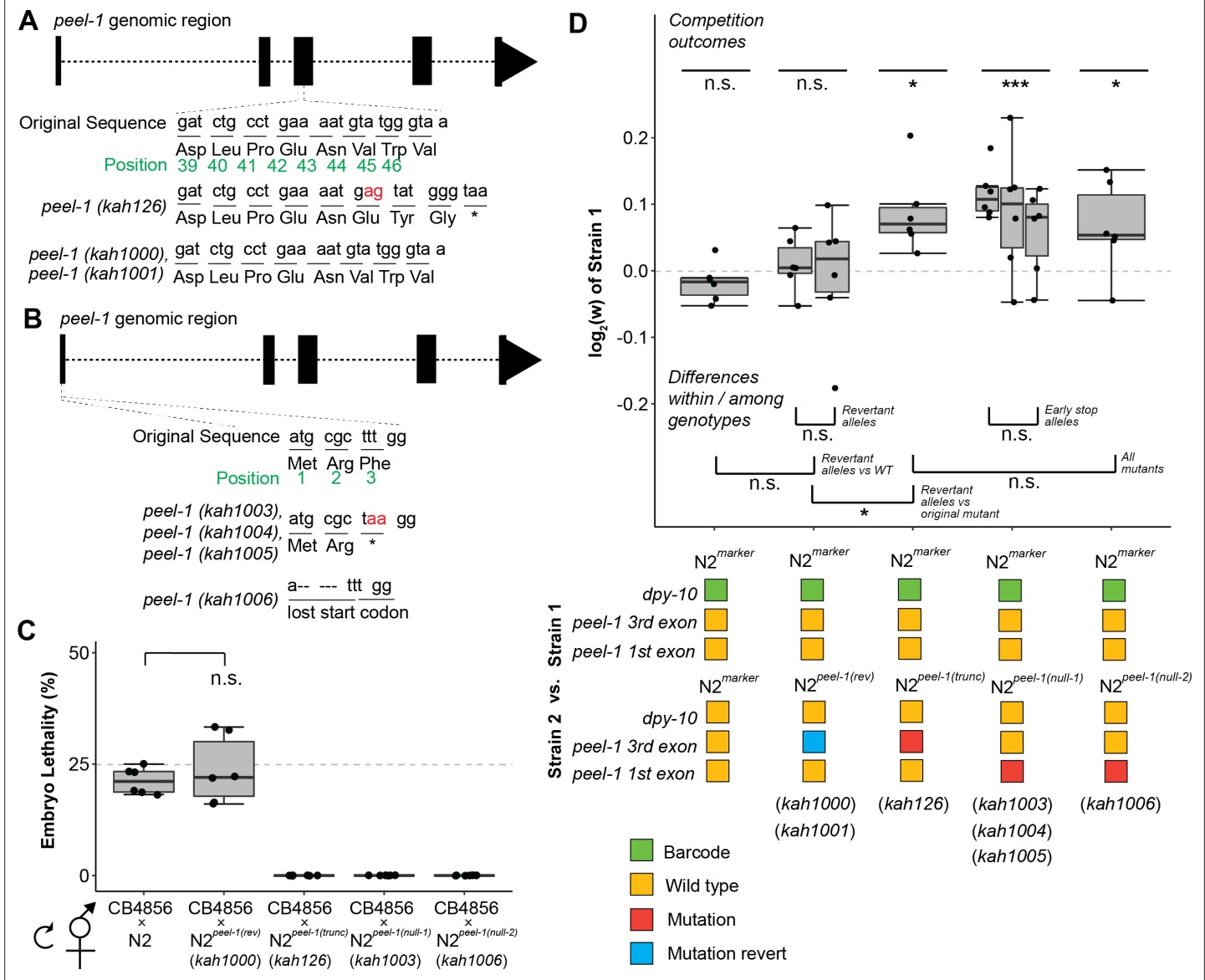

**Figure 4.** Additional *peel-1* alleles phenocopy the *peel-1* fitness effects. (**A**) Two independently derived *peel-1* revertant alleles (*kah1000*, *kah1001*) restore the original *peel-1* mutation (*kah126*) to wild-type. (**B**) Three independently derived *peel-1* alleles (*kah1003*, *kah1004*, *kah1005*) introduce an early stop in the third codon of the first exon; a 5 bp deletion in the first exon eliminates the start codon (*kah1006*). (**C**) Following crosses to CB4856, the progeny of selfed F1s confirm the expected toxin activity for these alleles: the *kah1000* revertant allele restores wild-type toxin activity, as N2^{peel-1 (rev)} selfed cross-progeny show ~25% lethality, similar to N2 (p=0.56, p=0.12, respectively, compared to null expectation of 25%); the *kah1003* and *kah1006* loss-of-function alleles eliminate toxin activity, as selfed cross-progeny from N2^{peel-1(null-1)} and N2^{peel-1(null-2)} produce zero dead embryos, the same as N2^{peel-1(trunc)} carrying the original *kah126* allele. (**D**) Strains carrying the revertant alleles (N2^{peel-1 (rev)}) show no fitness difference relative to the wild-type control but do show a fitness advantage relative to the original mutant with the truncated *peel-1* allele (N2^{peel-1(trunc)}), suggesting that the reversion edits restored *peel-1* function. Strains carrying the new *peel-1* null alleles (N2^{peel-1(null-1)}, N2^{peel-1(null-2)}) show a fitness disadvantage equivalent to the original mutant, N2^{peel-1(trunc)}, suggesting that all three mutant allele classes eliminate *peel-1* activity. Box plots show the central 50% of the dataset and the whiskers indicate 1.5× of the interquartile range; ***p<0.001, **p<0.01, and *p<0.05 by non-parametric analysis with correction for multiple tests (see 'Methods').

The online version of this article includes the following source data for figure 4:

**Source data 1.** Excel file containing source data for *Figure 4*.

PEEL-1 protein persists to adulthood and affects cellular function despite its presence at very low levels. Alternatively, PEEL-1 may induce long-lasting effects in adulthood via early developmental processes; notably, the toxic effect of PEEL-1 arises late in embryogenesis (*Seidel et al., 2011*). As the toxicity of PEEL-1 depends on the sex of the sperm donor, mediated in part by dosage (*Seidel et al.,*

*2011*), PEEL-1 may function as a mechanism to communicate parentage to the offspring. Additionally, it is possible that environmental conditions of the parent regulate PEEL-1 levels, which could also be communicated to the offspring.

## Conclusion

We have brought genomic editing and experimental evolution resources to bear on the study of a toxin-antidote element, addressing long-standing questions about their origin and maintenance in populations. We discovered that *peel-1* plays a role in individual fitness outside of its role as a toxin, affecting growth, fecundity, and fitness of non-hybrid genotypes, supporting recent arguments that non-selfish activity in inbred lineages may explain the prevalence of TA elements in non-obligate outcrossers (*Noble et al., 2021*; *Sweigart et al., 2019*). This work adds to the complicated nature of 'selfish' gene elements, similar to work in bacteria that has shown that TA elements can provide fitness benefits such as improved antibiotic resistance (*Bogati et al., 2022*). We hypothesize that other TA elements identified in *Caenorhabditis* species will also play roles outside of outcrossing, explaining how they can be retained in non-outcrossing populations.

## Methods
### Growth conditions

Strains were cultivated on agar plates seeded with *Escherichia coli* strain OP50 at 20°C (*Brenner, 1974*). The following strains were used in the study.

| Strain | Reference in text | Genotype | Comments |
|---|---|---|---|
| N2 | N2 | Wild-type reference | Isolated in Bristol, UK |
| CB4856 | CB4856 | Wild isolate | Isolated from a pineapple field on Oahu. |
| QX1198 | N2<sup>zeel-1;peel-1(CB4856)</sup> | qqIr5 [niDf9,CB4856>N2] I | *qqIr5* contains a 140–370 kb introgression from CB4856 into N2. |
| CX12311 | N2<sup>glb-5;npr-1(CB4856)</sup> | kyIR1[CB4856>N2] V; qgIR1 [CB4856>N2] X | *kyIR1* (V, CB4856>N2) is an introgression of the region surrounding *glb-5* from CB4856 into N2. *qgIR1* (X, CB4856>N2) is an introgression of the region surrounding *npr-1* from CB4856 into N2. Left breakpoint between 4,753,766 and 4,762,579. Right breakpoint between 4,882,488 and 4,885,498. |
| PTM229 | N2<sup>marker</sup> | dpy-10 (kah82) II | Silent mutation in *dpy-10*: Thr 90: acc ->act. |
| PTM377 | N2<sup>peel-1(trunc)</sup> | peel-1 (kah126) I | Original *peel-1* sequence: ATCTGCCTGAAAATGTATGGGTAAAT Mutated *peel-1* sequence: ATCTGCCTGAAAATGAGTATGGGTAAAT |
| PTM409 | N2<sup>peel-1(trunc);marker</sup> | peel-1(kah126) I; dpy-10 (kah82) II | PTM377 crossed with PTM229 to create this strain. |
| PTM1000 | N2<sup>peel-1(rev)</sup> | peel-1 (kah1000) I | *peel-1* reverted to wild type from PTM377 *peel-1 (kah126)* I. |
| PTM1001 | N2<sup>peel-1(rev)</sup> | peel-1 (kah1001) I | *peel-1* reverted to wild type from PTM377 *peel-1 (kah126)* I. |
| PTM1003 | N2<sup>peel-1(null1)</sup> | peel-1 (kah1003) I | *peel-1* stop codon introduced at the third amino acid. Original *peel-1* sequence: atgcgctttggtaagat Mutated *peel-1* sequence: atgcgctAAggtaagat |
| PTM1004 | N2<sup>peel-1(null1)</sup> | peel-1 (kah1004) I | *peel-1* stop codon introduced at the third amino acid. Original *peel-1* sequence: atgcgctttggtaagat Mutated *peel-1* sequence: atgcgctAAggtaagat |
| PTM1005 | N2<sup>peel-1(null1)</sup> | peel-1 (kah1005) I | *peel-1* stop codon introduced at the third amino acid. Original *peel-1* sequence: atgcgctttggtaagat Mutated *peel-1* sequence: atgcgctAAggtaagat |
| PTM1006 | N2<sup>peel-1(null2)</sup> | peel-1 (kah1006) I | *peel-1* 5 bp deletion in the first exon. Original *peel-1* sequence: atgcgctttggtaagat Mutated *peel-1* sequence: atttggtaagat |

CRISPR/Cas9 was used following a previously published co-conversion method to edit the target gene and *dpy-10* gene at the same time (*Arribere et al., 2014*). Generated strains are outcrossed to N2 more than three times before used for assay. Information on the N2 genome came from https://wormbase.org// and CeNDR (*Cook et al., 2017*). The following primers/sequences were used to create the CRISPR/Cas9 strains:

| Target allele | CRISPR/Cas9 Target site (19 bp) | Repairing oligo |
|---|---|---|
| peel-1 (kah126) I | gatctgcctgaaaatgtat | cagaaatctacatgtatcttgatctgcctgaaTGAgtatgggtaaatcggtttgcgcatgttattgctct |
| peel-1 (kah1003) I peel-1 (kah1004) I peel-1 (kah1005) I peel-1 (kah1006) I | gttttacaaggatgcgctt | ccgtcacaccaactgtggttttacaaggatgcgctaaggtaagattgttgtaatagcagaggaggcaaaggt |
| peel-1 (kah1000) I peel-1 (kah1001) I | tctgcctgaaaatgagtat | cagaaatctacatgtatcttgatctgcctgaaaatgtatgggtaaatcggtttgcgcatgttattgctct |

## Population dynamics prediction

All code to control population dynamics parameters and then plot the trajectories were stored at https://github.com/lijiang-long/TA_modeling (copy archived at *Long, 2023*). To calculate the allele frequency change at different frequencies of *zeel-1;peel-1,* the population is initiated with Hardy–Weinberg equilibrium such that the frequency of homozygous *zeel-1;peel-1* is the square of its allele frequency, and so on and so forth. The frequency of each genotype is updated each generation using the family-based toxin-antidote evolution dynamics in *Table 1*. This population is allowed to evolve five generations to deviate from Hardy–Weinberg equilibrium and reach the evolution trajectory of *zeel-1;peel-1*. The population evolves another generation, and the allele frequency change in this generation is used for plotting. To generate the heatmap where the frequency of *zeel-1;peel-1* after 1000 generations is plotted against varying outcrossing rate and fitness cost, the population is initiated with half *zeel-1;peel-1* allele. The genotype frequency is calculated assuming Hardy–Weinberg equilibrium. The population then evolves 1000 generations following *Table 1*. The final allele frequency of *zeel-1;peel-1* is then plotted on the heatmap.

## Competition assay to measure organism fitness

Competition experiments followed previous work (*Zhao et al., 2018*). Pairwise competition assays in *Figures 2 and 3* were done in parallel with the same start date. The competition assays in *Figure 4* were performed parallel with a different start date. All pairwise competition assays were performed on 9 cm NGM plates, seeded with OP50 bacteria, and stored at 4°C until 24 hr before use. At the beginning of the experiment, 10 L4 worms of each strain were transferred onto the same plate. This plate was then incubated at 20°C for 5 d. To propagate the next generation, a 1 cm agar chunk was transferred to a new 9 cm NGM plate. The old plate was then washed with 1 ml of M9 buffer to collect worms and stored at –80°C. Subsequently, this transfer and collection procedure was held every 3 d for a total of seven transfers. The genomic DNA from the first, third, fifth, and seventh transfer was isolated using Zymo 96-well DNA isolation kit (cat# D4071). Isolated genomic DNA was fragmented using EcoRI-HF by incubation at 37°C for 4 hr and purified using a Zymo 96-well DNA purification kit (cat# D4024). After purification, DNA concentrations were measured using Qubit DNA HS assay and adjusted to 1 ng/μl. To quantify the relative proportion of the two strains, a previously designed TaqMan probe was used targeting the *dpy-10* gene. After this, the DNA and TaqMan probe were mixed with the digital plate PCR (ddPCR) mix and processed through standard ddPCR procedures. The fractions of each strain were quantified using the BioRad QX200 machine with standard absolute quantification protocol. To estimate relative fitness, a linear regression model was applied to the DNA proportion data using the following equation with the assumption of one generation per transfer:

$$log\left(\frac{\frac{p\left(a\right)_0}{p\left(a\right)_t}-p\left(a\right)_0}{1-p\left(a\right)_0}\right) = \left(log\left(\frac{W_{aa}}{W_{AA}}\right)\right)t$$

where *p(a)* represents the relative allele proportion calculated from the ddPCR fluorescence signal and $W_{aa}$ and $W_{AA}$ represent the estimated fitnesses of the competing genotypes.

**Table 1.** A family-based model for the *zeel-1;peel-1* evolution dynamics.

| Family | Mating types | | | | Offspring genotype | | |
|---|---|---|---|---|---|---|---|
| | Sire | Dam | Frequency | Female fitness | PP | P+ | ++ |
| 1 | PP | PP | $X_{pp}X_{pp}k$ | 1-s | 1 | | |
| 2 | P+ | PP | $X_{p+}X_{pp}k$ | 1-s | 0.5 | 0.5 | |
| 3 | ++ | PP | $X_{++}X_{pp}k$ | 1-s | | 1 | |
| 4 | PP | P+ | $X_{pp}X_{p+}k$ | 1-hs | 0.5 | 0.5 | |
| 5 | P+ | P+ | $X_{p+}X_{p+}k$ | 1-hs | 0.25 | 0.5 | 0.25(1-t) |
| 6 | ++ | P+ | $X_{++}X_{p+}k$ | 1-hs | | 0.5 | 0.5 |
| 7 | PP | ++ | $X_{pp}X_{++}k$ | 1 | | 1 | |
| 8 | P+ | ++ | $X_{p+}X_{++}k$ | 1 | | 0.5 | 0.5(1-t) |
| 9 | ++ | ++ | $X_{++}X_{++}k$ | 1 | | | 1 |
| 10 | PP selfing | | $X_{pp}(1-k)$ | 1-s | 1 | | |
| 11 | *P*+selfing | | $X_{p+}(1-k)$ | 1-hs | 0.25 | 0.5 | 0.25(1-t) |
| 12 | ++selfing | | $X_{++}(1-k)$ | 1 | | | 1 |

Parameter X denotes the ratio of a certain genotype in a population. Genotype P denotes *zeel-1;peel-1* and +denotes 'no *zeel-1;peel-1*'. The parameter k specifies the outcrossing rate. When k = 1, there is complete outcrossing, and partial outcrossing is given by 0 < k < 1. The parameter s is the degree *zeel-1;peel-1* might reduce female fecundity. Dominance of the fecundity loss is defined by h. The parameter t models the paternal effect lethality. In the *zeel-1;peel-1* case, t is very close to 1.

## Embryo lethality assays

On day, a 10 cm NGM plate with plenty of gravid adults was bleached following standard protocol for each strain. Embryos were deposited to a 10 cm plates and incubated a 20°C. On day 2, 4–5 L3, young L4s hermaphrodites, and 8–10 CB4856 young L4 males were transferred to a mating plate for each of strain. Plates were incubated a 20°C. On day 5, adult hermaphrodites were singled on 6 cm plates (four plates per strain) and incubated at 20°C. on day 7, Plates were checked for males (F1) to determine if cross was successful. 20–30 F1 L4s from successfully crossed F0 herms were transferred to a 6 cm plates. Plates were incubated at 20°C. On day 8, four adult egg-laying adults (F1) were transferred to a 6 cm 'assay' plate for each replicate (six replicates per strain). Adults were on plates for 4 hr at room temperature. Adults were removed and plates incubated at 20°C for 18 hr. On days 9 and 12, dead embryos (F2) and adult worms were counted on days 9 and 12, respectively, and embryonic lethality was calculated for each replicate.

## Fecundity assays

Fecundity assays were performed at 20°C using 3 cm NGM plate seeded with 50 µl of OP50 bacteria with $OD_{600}$ of 2.0. The plates were allowed to dry overnight and stored at 4°C until 24 hr before use. At the beginning of the assay, six fourth larval stage (L4) worms were transferred to each assay plate. The worms were allowed to grow and lay eggs for the first 24 hr after the assay began before being transferred to a new plate. This process was repeated every 12 hr thereafter until animals ceased laying eggs. The number of eggs laid was counted using a standard dissecting microscope. This process is repeated every 12 hr thereafter until 100 hr or there is no egg on the new plate. The average fecundity was calculated by summing over all time points and dividing by the total number of worms in a single assay plate. While the data was initially collected every 12 hr, only the total fecundity was recorded for each assay.

## Growth rate assay

Growth rate assays were performed on standard NGM plates seeded with OP50 bacteria as previously described (*Large et al., 2016*). At the beginning of the assay, 10–20 adult worms were transferred onto an assay plate to lay eggs. After 2 hr, they were transferred off of the plate, leaving ~80 eggs

per plate. The plates were incubated for 72 hr at 20°C. At this point, the assay plate was mounted onto a video tracking camera and recorded for 1 min. The video clip was analyzed using a customized MATLAB script that tracks each animal and calculates the average size of each worm. The average size from each plate was then normalized by the average size of three N2 plates.

## Statistics

All hypothesis tests were performed using non-parametric analyses. One-sample comparisons to a null hypothesis value were assessed by the Wilcoxon test, and differences between pairs of samples were assessed by the Mann–Whitney $U$ test. For experiments with multiple comparisons, p-values were adjusted using the Benjamini and Hochberg method. For the fitness competitions testing replicate *peel-1* alleles against the wild-type control, we first evaluated each independently derived allele separately; as expected, the reversion alleles (*kah1000, kah1001*) showed no significant fitness differences while the early stop mutant alleles (*kah1003, kah1004, kah1005*) each showed a fitness disadvantage ($p < 0.05$ following correction for multiple tests). We also observed no significant differences among independently derived replicates within allele class, so we pooled replicate outcomes for further analyses (as reported in *Figure 4D*). For these tests, multiple comparison groups included the competition outcome tests and tests of replicate genotypes within allele class.

## Acknowledgements

We wish to acknowledge the core facilities at the Parker H Petit Institute for Bioengineering and Bioscience at the Georgia Institute of Technology for the use of their shared equipment, services, and expertise. Some strains were provided by the CGC, which is funded by NIH Office of Research Infrastructure Programs (P40 OD010440). We also thank the Kruglyak lab (UCLA) for strains. This research was supported in part through research cyberinfrastructure resources and services provided by the Partnership for an Advanced Computing Environment (PACE) at the Georgia Institute of Technology. This research was funded by NIH grant R35 GM119744 to ABP and NIH grant R35 GM139594 to PTM.

## Additional information

### Funding

| Funder | Grant reference number | Author |
| --- | --- | --- |
| National Institutes of Health | R35 GM139594 | Patrick T McGrath |
| National Institutes of Health | R35 GM119744 | Annalise B Paaby |

The funders had no role in study design, data collection and interpretation, or the decision to submit the work for publication.

### Author contributions

Lijiang Long, Conceptualization, Data curation, Formal analysis, Validation, Investigation, Visualization, Writing - original draft; Wen Xu, Data curation, Formal analysis, Methodology; Francisco Valencia, Data curation, Formal analysis; Annalise B Paaby, Conceptualization, Formal analysis, Supervision, Funding acquisition, Investigation, Visualization, Methodology, Writing - original draft, Project administration, Writing - review and editing; Patrick T McGrath, Conceptualization, Supervision, Funding acquisition, Validation, Investigation, Visualization, Methodology, Writing - original draft, Project administration

### Author ORCIDs

Lijiang Long (iD) http://orcid.org/0000-0002-9897-5900
Wen Xu (iD) http://orcid.org/0000-0003-2085-7223
Annalise B Paaby (iD) http://orcid.org/0000-0003-1422-047X
Patrick T McGrath (iD) http://orcid.org/0000-0002-1598-3746

### Decision letter and Author response

Decision letter https://doi.org/10.7554/eLife.81640.sa1

Author response https://doi.org/10.7554/eLife.81640.sa2

## Additional files

### Supplementary files
• MDAR checklist

### Data availability
All data generated or analysed during this study are included in the manuscript and supporting file. Source data files have been provided for all figures . Simulation code is included in a github: https://github.com/lijiang-long/TA_modeling (copy archived at *Long, 2023*).

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
