## [Editor Report]

This important work addresses how a selfish genetic element is maintained at intermediate frequencies in *C. elegans*. The evidence is convincing with both experimental and theoretical findings that tell us more about how these elements affect transmission in populations. Overall, the results of this study will be of broad interest to evolutionary biologists.

---

## [Decision Letter]

**Decision letter after peer review:**

Thank you for submitting your article "A Toxin-Antidote Selfish Element Increases Fitness of its Host" for consideration by *eLife*. Your article has been reviewed by 4 peer reviewers, including Erik C Andersen as Reviewing Editor and Reviewer #1, and the evaluation has been overseen by Christian Landry as the Senior Editor. The following individuals involved in the review of your submission have agreed to reveal their identity: Michael Ailion (Reviewer #3); Stefan Zdraljevic (Reviewer #4).

Essential revisions:

1) All reviewers noted the lack of controls for the peel-1 edited strain. I believe that backcrossing and whole-genome sequencing will not adequately address the concerns about the modest peel-1 effect and the single CRISPR-edited allele. The authors should either generate another peel-1 edited strain and test this new strain in the same experiments or add back peel-1 to the deleted strain to show that the modest peel-1 effect goes away.

2) Please update the statistical tests to address issues of multiple testing and non-normality.

3) The N2 data in Figures 2 and 3 are repeated. Please note that point.

4) Many of the other reviewer comments can be addressed by toning down some claims and providing more caveats and/or explanations. Reviewers made good suggestions for how to respond to these comments.

*Reviewer #3 (Recommendations for the authors):*

1. Though the mathematical modeling is interesting from a theoretical point of view, we feel that it oversells the rationale behind the experiments, setting up a "straw man" argument to knock down. Also, the modeling relies on rather high assumptions of the possible carrying cost of peel-1/zeel-1. For example, the modeling of the effect of outcrossing rate on peel-1/zeel-1 frequency assumes a selection coefficient of 0.35, which seems rather arbitrary and high. Where does this number come from? Is there any precedent for this high carrying cost? In our opinion, the idea that energy expenditure or leaky toxicity accounts for such a high carrying cost seems unlikely.

2. The two studies cited for "outcrossing rates typical for *C. elegans*" estimated vastly different outcrossing rates (~20% or ~1%). The model presented in Figure 1S2 specifically uses the lower estimates (0-2%), so the Sivasundar and Hey paper is miscited here. It is unclear whether there is a good rationale to go with the lower rate estimates. It would also be helpful to the reader if the value for these outcrossing rates is provided in the main text. Similarly, it would be helpful to provide in the main text the value for "all but the mildest carrying costs" (line 115).

3. The measurement of body size is unclear in the main text. Only when reading methods did we realize that body size is more of a proxy for growth rate rather than an end-point measurement of worm size. Please clarify this in the main text.

4. What is the temporal distribution of egg laying of the N2 and N2peel-1(null) strains? Based on how the data collection is described in the Methods, the authors should already have these data. Does egg-laying start at the same time in the two strains? The fact that strains carrying peel-1 grow faster but also apparently produce more sperm (which might slow them down) makes an analysis of this worthwhile, especially since fitness depends on when eggs are laid, not just how many. Some more characterization of this fitness trait seems appropriate and useful for beginning to understand how peel-1 may be increasing fitness. The original raw data for egg-laying with numbers of eggs laid at each time point should also be presented in the supplementary table rather than just the total amounts.

5. Line 65: the statement "similar elements have not been identified in obligate outcrossing Caenorhabditis nematodes" is somewhat misleading. TA elements may not have been identified in obligate outcrossing nematodes because of research bias since genetic experiments are easier to perform in non-obligate outcrossers and it is unclear that there have been extensive searches for TA elements in outcrossing nematodes. Furthermore, as the mathematical models in this study suggest, TA elements will spread quickly with an increasing rate of outcrossing. Since a TA element's non-fixation within a species has historically been a prerequisite for its discovery, the rapid TA element fixation that would generally occur in obligate outcrossers would make their identification more challenging.

6. Line 211-212: it is stated that this is the "first measurement of the fitness cost of a TA element to the host" and "first demonstration that a TA element can benefit the organism." These claims may be overstated. It has been previously shown in several cases that TA elements can provide fitness benefits to bacteria, such as improved antibiotic resistance (e.g. Bogati et al. 2022, PMID: 34570627). Also, the use of the phrase "fitness cost" in the first part of this sentence is confusing, since "cost" implies reduced fitness which is the opposite of what the paper shows for peel-1/zeel-1.

7. More details about the CRISPR protocol should be provided. It is unclear whether Cas9/sgRNAs were introduced as RNPs or plasmids (and at what concentrations). It is unclear how worms were screened for edits – the table has columns for primer sequences but these are blank. It is also unclear how many Dpy or Rol worms were screened and how many peel-1 or zeel-1 edited worms were found (the efficiency of CRISPR). The meaning of the shaded portion of the repairing oligo sequences in the table is not explained. Finally, it is not stated whether CRISPR-generated mutant strains were outcrossed.

8. The n values for fecundity assays are somewhat unclear. In the legends for Figure 2C and 3B, it says n=5 or n=6. Intuitively, most readers would interpret this as 5 or 6 worms, but according to the methods, it seems that this is 5 or 6 plates where each plate has a total of 6 worms whose average fecundity is calculated. This should be made clear in the figure legends. And though it says n=6 for the introgressed zeel-1 peel-1 strain, it seems that it is only 5 based on the source data in the supplementary table.

9. Line 185: it seems to be assumed that peel-1 arose before zeel-1, but the basis for this assumption is unclear.

10. Line 191 should cite Seidel et al. 2008 instead of Seidel et al. 2011.

11. The very right side of Figure 2C is cut off, making the rightmost column label incomplete. The legend of Figure 2 is also cut off at the bottom.

12. Figure 2B and 3C data are oversold visually due to the y-axis scale. There is only a 20-30 embryo difference but visually appears much more than that. It would be better to start the y-axis at 0.

13. Raw data for relative fitness experiments should be provided in a supplementary table.

14. The concluding sentence of the Results and Discussion section is unclear to us. What is meant by "outsized?"

15. The concluding sentence of the Conclusion would be better framed as a hypothesis rather than a belief.

16. How were outliers (mentioned in Figure 3 legend) identified/defined in the datasets?

17. Units should be added to column titles in the supplementary source data tables.

18. Line 243: the term "1k" is used but should be written out as "1000" to differentiate it from the "k" parameter used in the model.

19. Line 115: typo – "will likely to be" should read "will likely be".

20. Figure 1A typo: hermaphrodite.

*Reviewer #4 (Recommendations for the authors):*

1) It is unclear what, if any, experiments PTM573 (dpy-10 barcoded strain with the CB4856 hyper-divergent introgression) was used for. I am wondering why it is in the strain table.

1) Are the data for N2 presented in Figures 2B and 3C the same data? The boxplot distribution seems very similar by eye and without any data tables showing the actual values it is impossible for me to tell. Bacteria prep, agar prep, temperature, humidity levels, etc might affect the results on different days. So if the N2 values are from a single experiment, it might be best to repeat the experiment at the same time with the comparison strain. Ignore this comment if the data presented is actually from two separate experiments where the comparison strain was assayed simultaneously.

[Editors’ note: further revisions were suggested prior to acceptance, as described below.]

Thank you for resubmitting your work entitled "A Toxin-Antidote Selfish Element Increases Fitness of its Host" for further consideration by *eLife*. Your revised article has been evaluated by Christian Landry (Senior Editor) and a Reviewing Editor.

The manuscript has been improved but there are some remaining issues that need to be addressed, as outlined below:

The three reviewers appreciated all of the hard work and adjustments made to the manuscript. Overall, many of their comments can be addressed by additional explanations and addition of a few caveats. I have added short suggestions to the comments from each reviewer to help guide your edits.

*Reviewer #2 (Recommendations for the authors):*

In this revised version of their manuscript, Long et al. present further experimental evidence to support their claim that the peel-1 toxin, which is part of a paternal-effect toxin-antidote element in *C. elegans*, increases the competitive fitness of selfing hermaphrodites compared to those not carrying a copy. Several reviewers independently pointed out that to convincingly show this, the authors needed (1) to generate independent peel-1 knockout alleles that recapitulated the effect of the peel-1(kah126) mutant and (2) a rescue experiment, ideally a repair or reversion of the mutated allele back to WT to rule out any background effects. For this revision, the authors have generated additional strains and performed new competitive fitness assays to provide more convincing evidence that the fitness effect, albeit small, is indeed reproducible and caused by peel-1.

1. While the authors have addressed the major criticisms that focused on the genetics and reproducibility of the peel-1 mutant effect, there are some aspects of their fitness assays that remain unclear at the moment and would be important to clarify. The results shown in panel Figure 4D are central to the interpretation of the main claim of this paper, yet several questions come into mind upon close inspection, for instance:

1.1 Why is the data coming from independently generated alleles of the same genotype combined? Looking at the source data, it would appear that the authors replicated each pair-wise competition assay 6 times. However the data originating from independent alleles were combined in the final plots. As a result, the wt control and the original peel-1 mutant allele (kah126) have 6 replicates each, whereas other conditions, like the peel-1 "revertant" mutants and "new" peel-1 mutants have 12 and 18 replicates, respectively. I find it a bit troublesome that the fitness estimations are being done by aggregating independently derived lines (which goes against the original purpose of having independent alleles).

– The data for the different lines can be separated and reported.

1.2. Although it was brought up by a reviewer in the first round, it appears that there is no information available on the timing of the competitive fitness experiments. Were the experiments performed in parallel? And if not, when exactly? (this is important information for readers that should be available in the methods section/supplement).

– Please answer this question by adding text to the Methods.

2. I think it would be very important to check that the phenotype of the new peel-1 lines (new mutant alleles and revertant) is also consistent when measuring the total number of offspring laid by hermaphrodites (see Figure 3C)

– Without these data, the authors can mention that this trait was not measured for all of the peel-1 lines.

3. The authors may want to reconsider or justify the use of the term "biological role", for example as used in line 196

"These experiments strongly support a biological role for peel-1. "

I find the use of the term "biological role" very confusing in this context because it implies that the role of peel-1 as a toxin (or that of selfish elements in general) is "non- biological". Maybe the authors refer to a "physiological role" (not perfect either but more accurate)?

– Please edit to a different term. All genes have biological and physiological roles. Maybe "fitness-relevant role"?

Overall, this is a very exciting and interesting result and the manuscript has been greatly improved. Yet, many questions still remain open (which is OK if this is intended to be only a short report). How would a protein that is only expressed in sperm delay the development of hermaphrodites? Does this effect depend on zeel-1? Are the two opposing roles of peel-1 independent? (in other words, can one isolate mutants that abrogate toxicity but not the effect on fitness and vice versa). Does the benefit of carrying peel-1 evolved before or after the toxicity? Is there balancing selection and what is the selective force?

– Depending on space in the Discussion, some or all of these points can be addressed.

---

## [Author Response]

Essential revisions:1) All reviewers noted the lack of controls for the peel-1 edited strain. I believe that backcrossing and whole-genome sequencing will not adequately address the concerns about the modest peel-1 effect and the single CRISPR-edited allele. The authors should either generate another peel-1 edited strain and test this new strain in the same experiments or add back peel-1 to the deleted strain to show that the modest peel-1 effect goes away.2) Please update the statistical tests to address issues of multiple testing and non-normality.3) The N2 data in Figures 2 and 3 are repeated. Please note that point.4) Many of the other reviewer comments can be addressed by toning down some claims and providing more caveats and/or explanations. Reviewers made good suggestions for how to respond to these comments.

We appreciate the reviewers time and attention in reviewing our manuscript. We agree with their critiques and have addressed their major criticisms. Some of the main changes to the paper (marked in tracked changes):

1. There was significant worry that the conclusions relied on a single CRISPR-edited strain. Due to the small effect sizes, background effects or protein truncations could be responsible for the observed difference in fitness in the *peel-1* knockout strain. To address this, we created 6 new CRISPR-edited strains. Two of these strains reversed the knock out allele back to wild-type, which would leave the background mutations intact. These two new strains showed decreased fitness from the original *peel-1* allele, indicating that this *peel-1* allele was responsible for the fitness differences we observed vs. nonspecific background mutations. We also generated four additional loss of function alleles of *peel-1* in the first 3 amino acids of the protein. These loss-of-function mutations also showed decreased fitness from N2, further supporting our hypothesis that the fitness differences were from *peel-1* effects and not background mutations. These alleles also demonstrate that the fitness difference was not due to the expression of the translation of a partial PEEL-1 protein product. This data can be found in a new Figure 4.

2. We have updated the statistics to account for multiple testing and non-Guassian effects. For the fitness experiments, we modified the statistical test to determine if the fitness of one strain was significantly different from the second strain in a non-parametric way. This approach took advantage of the large number of comparisons we did for the competitions that had a small effect. The complete approach is detailed in the methods.

3. We have made a number of changes to address these specific comments of the reviewers:

a. We have updated nomenclature of genetic elements and strains as suggested by reviewers.

b. We have added references as suggested by reviewers.

c. We have deleted the line about discovery of these elements in obligate outcrossing nematodes (we agree that they many obligate male/female species have not been studied).

d. We have removed the effects of *zeel-1* from the manuscript (we agree that our initial conclusions about the role of *zeel-1* were incorrect).

e. We added a line to make it clear that our results suggest that additional natural polymorphisms linked to *peel-1* also affect laboratory fitness (as the NIL has a much stronger effect that the engineered *peel-1* strains).

f. We have removed the Sivasundar reference and added a review that further supports the low outcrossing rate. We have made explicit the carrying costs as suggested by reviewer 3.

g. We have removed PTM573 from the paper.

h. We have indicated where the data was shared between different figures.

4. We unfortunately do not have the detailed egg-laying rates for each strain to further analyze. While the experiments suggested by Reviewer 3 would be interesting, we believe they are outside of the scope of the paper (short report). Re Reviewer 4’s comments on starvation, differences in their response also could play a role in the laboratory fitness. Teasing out the exact contribution of egg-laying, growth, dauer formation, starvation, etc would be laborious.[Editors’ note: what follows is the authors’ response to the second round of review.]

Reviewer #2 (Recommendations for the authors):1. While the authors have addressed the major criticisms that focused on the genetics and reproducibility of the peel-1 mutant effect, there are some aspects of their fitness assays that remain unclear at the moment and would be important to clarify. The results shown in panel Figure 4D are central to the interpretation of the main claim of this paper, yet several questions come into mind upon close inspection, for instance:1.1 Why is the data coming from independently generated alleles of the same genotype combined? Looking at the source data, it would appear that the authors replicated each pair-wise competition assay 6 times. However the data originating from independent alleles were combined in the final plots. As a result, the wt control and the original peel-1 mutant allele (kah126) have 6 replicates each, whereas other conditions, like the peel-1 "revertant" mutants and "new" peel-1 mutants have 12 and 18 replicates, respectively. I find it a bit troublesome that the fitness estimations are being done by aggregating independently derived lines (which goes against the original purpose of having independent alleles).– The data for the different lines can be separated and reported.

We revised our presentation of these results to include comparisons within and among the independently derived alleles. Our results now show that all independently derived but identical alleles are equivalent to each other in the competition assays, and further, that all three allele classes of the peel-1 mutants are equivalent to each other. Figure 4D has been updated to display the replicate lines separately, and the text has been updated to describe our analysis in finer detail.

1.2. Although it was brought up by a reviewer in the first round, it appears that there is no information available on the timing of the competitive fitness experiments. Were the experiments performed in parallel? And if not, when exactly? (this is important information for readers that should be available in the methods section/supplement).– Please answer this question by adding text to the Methods.

Added ‘Pairwise competition assays in figure 2 and figure 3 were done in parallel with the same start date. The competition assays in figure 4 were performed parallel with a different start date.’ to the method section.

2. I think it would be very important to check that the phenotype of the new peel-1 lines (new mutant alleles and revertant) is also consistent when measuring the total number of offspring laid by hermaphrodites (see Figure 3C)– Without these data, the authors can mention that this trait was not measured for all of the peel-1 lines.

We added this line (195).

3. The authors may want to reconsider or justify the use of the term "biological role", for example as used in line 196"These experiments strongly support a biological role for peel-1. "I find the use of the term "biological role" very confusing in this context because it implies that the role of peel-1 as a toxin (or that of selfish elements in general) is "non- biological". Maybe the authors refer to a "physiological role" (not perfect either but more accurate)?– Please edit to a different term. All genes have biological and physiological roles. Maybe "fitness-relevant role"?

Changed in this line and in the conclusion.

Overall, this is a very exciting and interesting result and the manuscript has been greatly improved. Yet, many questions still remain open (which is OK if this is intended to be only a short report). How would a protein that is only expressed in sperm delay the development of hermaphrodites? Does this effect depend on zeel-1? Are the two opposing roles of peel-1 independent? (in other words, can one isolate mutants that abrogate toxicity but not the effect on fitness and vice versa). Does the benefit of carrying peel-1 evolved before or after the toxicity? Is there balancing selection and what is the selective force?– Depending on space in the Discussion, some or all of these points can be addressed.

We added an additional paragraph at the end of the Results to address these points.